



# Optimal scheduling of the next preventive maintenance activity for a wind farm

Quanjiang Yu , Michael Patriksson , and Serik Sagitov

Department of Mathematical Sciences, Chalmers University of Technology and University of Gothenburg, SE-42 196 Gothenburg, Sweden

**Correspondence:** Quanjiang Yu (yuqu@chalmers.se)

**Abstract.** Global warming has been attributed to increased greenhouse gas emission concentrations in the atmosphere through the burning of fossil fuels. Renewable energy, as an alternative, is capable of displacing energy from fossil fuels. Wind power is abundant, renewable, and produces almost no greenhouse gas during operation. A large part of the cost of operations is due to the cost of maintaining the wind power equipment, especially for offshore wind farms. How to reduce the maintenance cost

is what this article focus on.

This article presents a binary linear optimisation model whose solution may suggest wind turbine owners which components, and when, should undergo the next preventive maintenance (PM). The scheduling strategy takes into account eventual failure events of the multi-component system, in that after the failed system is repaired, the previously scheduled PM plan should be updated treating the restored components to be as good as new.

The optimisation model NextPM is tested through three numerical case studies. The first study addresses the illuminating case of a single component system. The second study analyses the case of seasonal variations of set-up costs, as compared to the constant set-up cost setting. Among other things, this analysis reveals a dramatic cost reduction achieved by the NextPM model as compared to the the pure CM strategy. In these two case studies, the cost are reduced by around $35\%$. The third case study compares the NextPM model with another optimisation model preventive maintenance scheduling problem with interval

costs(PMSPIC) which was the major source of inspiration for this article. This comparison demonstrates that the NextPM model is accurate and much more effective.

In conclusion, the NextPM model is both accurate and fast to solve. The algorithm stemming from the proposed model can be used as a key module in a maintenance scheduling app.

## 1  Introduction

Wind energy is one of the lowest-priced renewable energy technologies available today; see Lazard (2017). A large part of the cost of operations is due to the cost of maintaining the wind power equipment, especially for offshore wind farms. To further reduce the maintenance cost, one can improve the design of the components, making them more reliable. One can also reduce the maintenance costs by means of an improved scheduling of the component replacements. The latter task is the main motivation for this paper, which proposes an optimisation model dealing with a single wind turbine, or more generally with a





single multi-component system. The suggested approach has a straightforward extension to a group of several multi-component systems (a farm of wind turbines) with possibly different parameters.

Typically, a maintenance model distinguishes between a corrective maintenance (CM) event, when a component should be attended after it breaks down, and a preventive maintenance (PM) event, when one or several components are renewed before they break down; see the recent survey Lee and Cha (2016). An optimal PM scheduling is aimed at reducing the lost production

due to the down-time caused by CM events.

There are multitude of papers devoted to the optimal PM scheduling for multi-component systems, see Werbińska-Wojciechowska et al. (2019). The article Jafari et al. (2018) proposes a joint optimization of the maintenance policy and the inspection interval for a multi-unit series system with economic dependence. It develops a model and algorithm that can be used to determine an optimal maintenance policy for a multi-component system to optimize the maintenance cost, where one unit is subject to

condition monitoring, while just the age information is available for the other unit, and which has a general distribution. Tian et al. (2014) developed a method to quantify uncertainty of the remaining life length by considering the prediction accuracy improvement, and an effective condition based maintenance optimization approach to optimize the maintenance schedule.

The article Sarker and Faiz (2016) looks at opportunistic maintenance which is a special kind of preventive maintenance. When one component breaks down, since the maintenance personal need to go there and maintain them, they may as well

maintain some other components to save some logistic costs. This is extremely beneficial for offshore wind farms, due to the large set-up costs.

In Moghaddam and Usher (2011), optimization models are developed to determine the optimal preventive maintenance and replacement schedules in repairable and maintainable systems. They showed that if there is a fixed set-up cost, it forces preventive maintenance and replacement activities to occur at the same time. However, their models are nonlinear, which makes

them computationally hard to solve.

The PMSPIC (Preventive Maintenance Scheduling Problem with Interval Costs) model from Gustavsson et al. (2014) was the major inspiration for this work. The main feature of the PMSPIC model is a rescheduling step characterised by a cost function of the planned PM which depends on the time between two consecutive PM activities.

In Section 2 the framework of the optimisation model is introduced for a multi-component system in a discrete time setting

$t = 0, 1, \ldots, T$, where the unit of time can be a day or a month or a year, depending on the concrete situation. It is assumed that at time 0 all the components of the system were new and that the system lifespan is $T$. In the same Section 2 the main result is summarised as Algorithm 1, producing an optimal PM scheduling for the time period $[s, T]$, starting at some given time $s \in [0, T-1]$.

The key ingredient of Algorithm 1, the NextPM algorithm, is carefully described in Section 3, where the key differences

between the approach in this article and that of Gustavsson et al. (2014) are also clearly specified.

Section 4 contains several numerical studies that demonstrate the flexibility of the approach in this article, its accuracy and effectiveness, which makes it relevant as a part of a future app for PM scheduling for farms of wind turbines.

For the motivated reader, complete formal presentations of the linear optimisation models from Section 3.1 and Section 3.4 are presented in Appendix A and Appendix B, respectively.



## 2 Optimal rescheduling algorithm

Consider a system composed of $n$ components characterised by different life length distributions. For the component $j$, it is assumed that its total life length is $L_j$, without any maintenance, has a Weibull distribution with parameters $(\alpha_j, \beta_j)$, so that the corresponding survival function has the following parametric form

$$\mathrm{P}(L_j > t) = e^{-(\frac{t}{\alpha_j})^{\beta_j}}, \quad t \geq 0, \quad j = 1, \ldots, n; \tag{1}$$

see Guo et al. (2009) concerning the use of the Weibull distribution for the modelling of multi-component systems. The means and variances of the component life lengths are computed as

$$\mu_j = \alpha_j \Gamma(1 + \tfrac{1}{\beta_j}), \quad \sigma_j^2 = \alpha_j^2 \Gamma(1 + \tfrac{2}{\beta_j}) - \mu_j^2, \quad j = 1, \ldots, n. \tag{2}$$

Besides the component survival parameters $(\alpha_j, \beta_j)$, the optimisation model of a multi-component system requires the following parameters describing the various maintenance costs:

    $d_t$, the time-dependent set-up cost for either a PM or CM activity,

    $b_j$, the CM cost of the component $j$,

    $c_j$, the PM cost of the component $j$,

where time $t = 0, \ldots, T$ is discrete and $j = 1, \ldots, n$. To summarise, the full set of the model parameters is $\{d_1, \ldots, d_T, (\alpha_1, \beta_1, b_1, c_1), \ldots, (\alpha_n, \beta_n, b_n, c_n), \lambda\}$, including an extra parameter $\lambda$ introduced in Section 3.2 by formula (11). Notice that the definition of $d_t$ covers the continuous range of the variable $t$ by setting $d_t = d_{\lceil t \rceil}$.

Suppose that the multi-component system is observed at some discrete time $s \in [0, T-1]$, and the last maintenance times are $t_j \in [0, s]$ for each of the components $j = 1, \ldots, n$, so that at the time $s$ the $n$ components have the effective ages $(s - t_1, \ldots, s - t_n)$. The NextPM optimisation model described in Section 3, has the input $(t_1, \ldots, t_n, s, r)$, where $r \in [s+1, T]$ is the end of the current planning period. The output of NextPM is a PM plan specifying the time $\tau \in [s+1, r+1]$ of the next PM event, as well as which of the components $\mathcal{P} \subset \{1, \ldots, n\}$ should be maintained at the time $\tau$. The $\tau = r+1$ implies $\mathcal{P} = \emptyset$ which means that no PM should be scheduled during the planning period $[s+1, r]$.

The NextPM is the key module of the following algorithm for PM scheduling until the time $T$ when the whole system is expected to be dismantled. The algoritm relies on a rescheduling procedure, where each NextPM step covering $r - s$ units of the planning time is accompanied by a NextOM module. The latter is a modification of NextPM, see Section 3.4, which addresses the possibility of a component failure before the planned PM, followed by opportunistic maintenance (OM) activities.





---

**Algorithm 1** Optimal rescheduling algorithm

Input $t_1, \ldots, t_n, s, r$

Start

        Solve NextPM$\{t_1, \ldots, t_n, s, r\}$

        Output $\tau$, $\mathcal{P}$, where $\mathcal{P} \subset \{1, \ldots, n\}$

        If $\tau < T$

                If a failure during the period $(s, \tau]$ damages component $i$ at time $u_i$

                        Set $u := \lfloor u_i \rfloor$

                        Solve NextOM$\{i, t_1, \ldots, t_n, u\}$

                        Output $\mathcal{O} \subset \{1, \ldots, n\}$

                        Perform CM of component $i$ at time $u + 1$

                        Perform PM of each component $j \in \mathcal{O}$ at time $u + 1$

                        Update $r := \min(u + 1 + r - s, T)$, $s := u + 1$

                        Update $t_j := u + 1$, $j \in \mathcal{O} \cup \{i\}$

                Else

                        Perform PM of each component $j \in \mathcal{P}$ at time $\tau$

                        Update $r := \min(\tau + r - s, T)$, $s := \tau$, $t_j := s$, $j \in \mathcal{P}$

                End

                Go to Start

        Else

                Stop

        End

---

Comments:

    $\mathcal{P}$ is the set of components that should undergo PM at time $\tau$,

    $\mathcal{O}$ is the set of components that should undergo OM at time $u + 1$.

## 3   An optimal plan for the next preventive maintenance

This section sets up the optimisation model NextPM, which is the key ingredient of Algorithm 1 summarised in Section 2. The purpose of the NextPM model is to produce an optimal PM plan for the period $[s + 1, r]$, where the planning timespan $r - s$ is chosen so that it is reasonable to expect at most one PM event during time $r - s$.





### 3.1 NextPM model

For a given planning period $[s+1, r] \subset [0, T]$, an $(s, r)$-*plan* is defined as any set of vectors $(\boldsymbol{z}, \boldsymbol{x}^1, \ldots, \boldsymbol{x}^n)$ whose components are vectors

$$\boldsymbol{z} = (z_{s+1}, \ldots, z_{r+1}), \qquad \boldsymbol{x}^j = (x_{s+1}^j, \ldots, x_{r+1}^j), \quad j = 1, \ldots, n,$$

with binary coordinates $z_t, x_t^j \in \{0, 1\}$, which satisfy the following linear conditions:

$$\sum_{t=s+1}^{r+1} x_t^j = 1, \quad j = 1, \ldots, n, \tag{3}$$

$$x_t^j \leq z_t, \quad t = s+1, \ldots, r+1, \ j = 1, \ldots, n. \tag{4}$$

For $t = s+1, \ldots r$, the equality $x_t^j = 1$ means that

the optimal schedule is to *tentatively* plan to perform a PM of the component $j$ at the time step $t$: whenever a failure of the component occurs during the period $[s+1, t]$, the plan requires rescheduling of the next PM.

Likewise, $z_t = 1$ means that the optimal schedule is to tentatively plan to perform maintenance of at least one of the components at the time step $t$. Furthermore, $x_{r+1}^j = 1$ if the optimal schedule is not plan to perform maintenance for the component $j$ during the time period $[s+1, r]$. The equality $z_{r+1} = 1$ means that the optimal schedule does not plan any maintenance activity during the time period $[s+1, r]$.

The NextPM optimisation model is built around the objective function

$$f(\boldsymbol{z}, \boldsymbol{x}^1, \ldots, \boldsymbol{x}^n) = \sum_{t=s+1}^{r+1} \frac{1}{t-s} \left( d_t z_t + c_{s,t}^1 x_t^1 + \ldots + c_{s,t}^n x_t^n \right), \tag{5}$$

where $c_{s,t}^j$ is defined in Section 3.2. Since $d_t z_t$ stands for the set-up cost and the sum $\sum_{j=1}^{n} c_{s,t}^j x_t^j$ gives the total maintenance cost, the objective function (5) should be viewed as the *time-average* maintenance cost per time unit according to the $(s, t)$-plan $(\boldsymbol{z}, \boldsymbol{x}^1, \ldots, \boldsymbol{x}^n)$.

Let $(\bar{\boldsymbol{z}}, \bar{\boldsymbol{x}})$ be the solution to the linear optimisation problem aimed to

$$\text{minimize} \quad f(\boldsymbol{z}, \boldsymbol{x}^1, \ldots, \boldsymbol{x}^n), \tag{6}$$

over all $(s, t)$-plans subject to the linear constraints

$$D_{s,t}^j x_t^j \geq 0, \quad t = s+1, \ldots, r, \ j = 1, \ldots, n, \tag{7}$$

where $D_{s,t}^j$ is defined in Section 3.3 as the PM benefit for the component $j$ at time $t$. Then the output of the NextPM algorithm $(\tau, \mathcal{N})$ recommends the time of the next PM

$$\tau = \min_j \{\arg\max_t \bar{x}_t^j\}$$





along with the set of the components that should undergo the maintenance activities

$$
\mathcal{N} = \begin{cases} \{j : \bar{x}_\tau^j = 1, \ j = 1, \dots, n\} & \text{if } \tau \le r, \\ \emptyset & \text{if } \tau = r + 1. \end{cases}
$$

## 3.2 Definition of $c_{s,t}^j$

Here we deal with the term $c_{s,t}^j$ appearing in the the objective function (5) of the optimisation model NextPM. The main idea
is to define $c_{s,t}^j$ as the fixed PM cost $c_j$ plus the expected additional costs due to eventual failures of the component $j$ occurring
prior to the planned PM activity at time $t$.

To this end, consider $n$ independent sequences of renewal times with a delay by letting $U_{s,0}^j = s$,

$$
U_{s,1}^j = t_j + L_{1j}, \quad L_{1j} \overset{d}{=} \{L_j | L_j > s - t_j\}, \tag{8}
$$

where $\overset{d}{=}$ means equality in distribution (conditional distribution in the above formula), and

$$
U_{s,i+1}^j = U_{s,i}^j + L_{ij}, \quad L_{ij} \overset{d}{=} L_j, \quad \text{for } i = 2, 3, \dots, \tag{9}
$$

assuming that the random variables $(L_{ij})$ are mutually independent. Notice that in the important particular case $s = 0$, this
definition simplifies, so that for each $j$, the sequence $\{U_{0,i}^j\}_{i \ge 0}$ describes a renewal process without a delay.

Treating $U_{s,1}^j, U_{s,2}^j, \dots$ as the consecutive failure times of the component $j$, put

$$
c_{s,t}^j := c_j + \mathrm{E}\left( \sum_{i=1}^{\infty} 1_{\{U_{s,i}^j \le t\}} G_j(U_{s,i-1}^j, L_{ij}, t - s) \right), \tag{10}
$$

where the cost functions

$$
G_j(s, u, t) = b_j + d_{s+u} - \left(\frac{u}{t}\right)^\lambda (c_j + d_{s+t}), \quad 0 \le u \le t, \tag{11}
$$

involve a new parameter $\lambda > 0$ assumed to be independent of $j = 1, \dots, n$. The definition of the cost function (11) further
develops the idea of Section 5.1 in Gustavsson et al. (2014); see Section 3.5 below. It describes the additional cost incurred at
the failure time $s + u$, taking place between the starting time $s$ and the time $s + t$ scheduled for the next PM. If $u$ is close to 0,
then the failure at time $s + u$ will not change the PM plan, implying that the additional cost

$$
G_j(s, 0, t) = b_j + d_s \tag{12}
$$

is the sum of the CM cost $b_j$ and the set-up cost $d_s$ at time $s$. On the other hand, if $u$ is close to $t$, then the additional cost

$$
G_j(s, t, t) = b_j - c_j \tag{13}
$$

is simply the difference between the CM and PM costs. The expression in the righthand side of (11) represents an intermediate
additional cost, where the parameter $\lambda$ evaluates to what extent the proximity of $u$ to $t$ reduces the planned PM costs.



### 3.3 Definition of $D_{s,t}^{j}$

The constraint (7) arises as a check-up step to ensure that a suggested PM at time $t$ brings some benefit, as compared to a simple strategy when no PM is performed. With the PM-free strategy, the total maintenance cost (including set-up costs) for the component $j$ during the period $[s,T]$ would be

$$\mathrm{E}\left[\sum_{i=1}^{\infty} 1_{\{U_{s,i}^{j} \leq T\}}\left(b_{j}+d_{U_{s,i}^{j}}\right)\right].$$

Alternatively, if the plan is to perform a PM for the component $j$ at time $t$, and then to perform replacements of the component $j$ whenever it breaks down, then the total cost would be

$$c_{s,t}^{j}+\mathrm{E}\left[\sum_{i=1}^{\infty} 1_{\{t+U_{0,i}^{j} \leq T\}}\left(b_{j}+d_{t+U_{0,i}^{j}}\right)\right].$$

Taking into account the difference between these two total costs

$$D_{s,t}^{j}=\mathrm{E}\left[\sum_{i=1}^{\infty} 1_{\{U_{s,i}^{j} \leq T\}}\left(b_{j}+d_{U_{s,i}^{j}}\right)\right]-c_{s,t}^{j}-\mathrm{E}\left[\sum_{i=1}^{\infty} 1_{\{t+U_{0,i}^{j} \leq T\}}\left(b_{j}+d_{t+U_{0,i}^{j}}\right)\right], \tag{14}$$

we conclude that the planned PM of the component $j$ at time $t$ is justified only if $D_{s,t}^{j} \geq 0$.

### 3.4 NextOM model

The NextOM part of Algorithm 1 is a specialised version of the NextPM part described below in terms of a given input vector

$$(i,t_{1},\ldots,t_{i-1},t_{i+1},\ldots,t_{n},s).$$

Here, $s \in [0,T]$ and $i$ is the label of the component whose failure at some time during $[s,s+1)$ has triggered the OM planning step. For a pair $\{s,i\}$, an $\{s,i\}$-*plan* is any set of vectors $(\boldsymbol{z},\boldsymbol{x}^{1},\ldots,\boldsymbol{x}^{n})$ whose components are two-dimensional vectors

$$\boldsymbol{z}=(z_{s+1},z_{s+2}), \qquad \boldsymbol{x}^{j}=(x_{s+1}^{j},x_{s+2}^{j}), \quad j=1,\ldots,n, \tag{15}$$

with binary coordinates $z_{t}$, $x_{t}^{j} \in \{0,1\}$ satisfying the following linear conditions

$$\sum_{t=s+1}^{s+2} x_{t}^{j}=1, \quad j=1,\ldots,n, \tag{16}$$

$$x_{s+1}^{(i)}=1, \tag{17}$$

$$z_{t} \geq x_{t}^{j}, \quad t=s+1,s+2, \quad j=1,\ldots,n. \tag{18}$$

Observe that necessarily, $z_{s+1}=1$.

The NextOM optimisation model uses a modified objective function

$$f_{i}(\boldsymbol{z},\boldsymbol{x}^{1},\ldots,\boldsymbol{x}^{n})=\sum_{t=s+1}^{s+2}\frac{1}{t-s}\left(d_{t}z_{t}+\sum_{j \neq i}c_{s,t}^{j}x_{t}^{j}\right), \tag{19}$$





where $c_{s,t}^j$ is defined in Section 3.2. Let $(\bar{z}, \bar{x})$ be the solution to the linear optimisation problem to

$$\text{minimise} \quad f_i(z, x^1, \ldots, x^n) \tag{20}$$

over all $\{s, i\}$-plans subject to the linear constraint

$$D_{s,s+1}^j x_{s+1}^j \geq 0, \quad j = 1, \ldots, i-1, i+1, \ldots, n, \tag{21}$$

where $D_{s,t}^j$ is defined in Section 3.3. The output of the NextOM is given by the set

$$\mathcal{O} = \{j : \bar{x}_\tau^j = 1, \ j = 1,, \ldots, i-1, i+1, \ldots, n\},$$

consisting of the labels of the components which will be opportunistically maintained along with the component $i$ undergoing a CM activity.

### 3.5 Comparison with the PMSPIC optimisation model

The optimisation model PMSPIC of Gustavsson et al. (2014) is presented here in terms similar to the current setting. It is compared to the optimisation model presented in this article in the particular case when $s = 0$ and the set-up costs $d_t \equiv d$ are constant over time.

For the planning period $[0, T]$ of the PMSPIC model, define the set of paired time points

$$\mathcal{I} = \{(u, t) : 0 \leq u < t \leq T + 1\},$$

and call an $\mathcal{I}$-plan any vector $(z, x^1, \ldots, x^n)$ composed by a vector and $n$ triangular arrays

$$z = (z_1, \ldots, z_T), \quad x^j = \{x_{ut}^j, \ (u, t) \in \mathcal{I}\}, \ j = 1, \ldots, n.$$

It is assumed that the binary components $z_t, x_{ut}^j \in \{0, 1\}$, satisfy the following linear conditions

$$z_t \geq \sum_{u=s}^{t-1} x_{ut}^j, \quad t = 1, \ldots T, \ j = 1, \ldots, n, \tag{22}$$

$$\sum_{t=1}^{T+1} x_{st}^j = 1, \quad j = 1, \ldots, n, \tag{23}$$

$$\sum_{u=0}^{t-1} x_{ut}^j = \sum_{v=t+1}^{T} x_{tv}^j, \quad t = 1, \ldots T+1, \ j = 1, \ldots, n. \tag{24}$$

For $(u, t) \in \mathcal{I}$, the equality $x_{ut}^j = 1$ means that according to the $\mathcal{I}$-plan, the component $j$ will be maintained both at the time step $u$ and time step $t$ but not in between. The equality $z_t = 1$ means that it is planned to perform a maintenance of at least one of the components at the time step $t$. The constraint (24) is the counterpart of the flow balance constraint from Fulkerson (1966).



The PMSPIC model minimises the objective function

$$F(\boldsymbol{z}, \boldsymbol{x}^1, \ldots, \boldsymbol{x}^n) = \sum_{t=1}^{T} d_t z_t + \sum_{(u,t) \in \mathcal{I}} \sum_{j=1}^{n} c_{t-u}^{j} x_{ut}^{j}, \tag{25}$$

representing the *total* maintenance cost of the $\mathcal{I}$-plan $(\boldsymbol{z}, \boldsymbol{x}^1, \ldots, \boldsymbol{x}^n)$. Here the term $c_t^j$ given by

$$c_t^j = c_j + \mathrm{E}\left( \sum_{i=1}^{\infty} 1_{\{U_{0,i}^j \leq t\}} g_j(U_{0,i}^j, t) \right), \tag{26}$$

where

$$g_j(u, t) = b_j + d - \left(\tfrac{u}{t}\right)^{\lambda}(c_j + d) \tag{27}$$

should be compared with the term $c_{0,t}^j$ defined by (10) and (11), which in the particular case of $s = t_1 = \ldots = t_n = 0$ and $d_t \equiv d$ takes the form

$$c_{0,t}^j = c_j + \mathrm{E}\left( \sum_{i=1}^{\infty} 1_{\{U_{0,i}^j \leq t\}} g_j(L_{ij}, t) \right). \tag{28}$$

Comparison of the two expressions for $c_t^j$ and $c_{0,t}^j$, reveals that the key difference is between the terms $g_j(U_{0,i}^j, t)$ and $g_j(L_{ij}, t)$. The formula $g_j(U_{0,i}^j, t)$ for $i \geq 2$ is not compatible with the meaning of the cost function $g_j(u, t)$ explained earlier for (11). Indeed, the term $g_j(U_{0,i}^j, t)$ assumes that the component $j$ has age $U_{0,i}^j$, while actually it is supposed to be restored at the time $U_{0,i-1}^j$ of the previous failure.

## 4   Numerical studies

The three case studies presented in this section deal with a wind turbine as an example of the multi-component system. They are all based on the parameter values taken from the paper Tian et al. (2011), see Table 1, where the cost unit is 1000 USD and

| Component | $j$ | CM cost $b_j$ (1000 $) | PM cost $c_j$ (1000 $) | $\beta_j$ | $\alpha_j$ (months) | $\mu_j(months)$ |
|---|---|---|---|---|---|---|
| Rotor | 1 | 162 | 36.75 | 3 | 100 | 89.9 |
| Main bearing | 2 | 110 | 23.75 | 2 | 125 | 110.8 |
| Gearbox | 3 | 202 | 46.75 | 3 | 80 | 71.4 |
| Generator | 4 | 150 | 33.75 | 2 | 110 | 97.5 |

**Table 1.** Key parameters for a four-component system.

the time unit is 1 month. The lifetime of the wind turbine is assumed to be 20 years, which is the typical case in the industry now, according to Ziegler et al. (2018). This implies the value $T = 240$ [months]. It is assumed that

$s = 0$ which implies that all four components initially are as good as new,





$r = 80$, see Section 4.1 for motivation,

$\lambda = 3$ was deemed to be relevant based on the analysis of computer simulations which is not reported here.

All computational tests are performed on an Intel 2.40 GHz dual core Windows PC with 16 GB RAM. The mathematical optimisation models are implemented in AMPL IDE (version 3.5); the model components (10) and (7) are calculated by Matlab

(version R2015b), and then the optimisation problems are solved using CPLEX (version 12.8).

### 4.1 Study 1: a single-component system

If $n = 1$, $d_t \equiv d$, and $s = 0$, the objective function (5) takes the form

$$f(\boldsymbol{x}) = \sum_{t=1}^{r+1} a_t x_t, \quad a_t = \frac{d + c_t}{t}, \tag{29}$$

where given a sequence of independent random variables $L_i \overset{d}{=} L$ with $L$ having a Weibull $(\alpha, \beta)$ distribution,

$$c_t = c + \mathrm{E}\left( \sum_{i=1}^{\infty} 1_{\{L_1 + \ldots + L_i \leq t\}} \left[ b + d - (\tfrac{L_i}{t})^{\lambda}(c + d) \right] \right). \tag{30}$$

It turns out that in the current setting, the constraint (7) can effectively be disregarded so that the optimal PM time $\tau$ is obtained by minimising the objective function (5), which is equivalent to minimising $a_t$ over $t = 1, \ldots, r+1$.

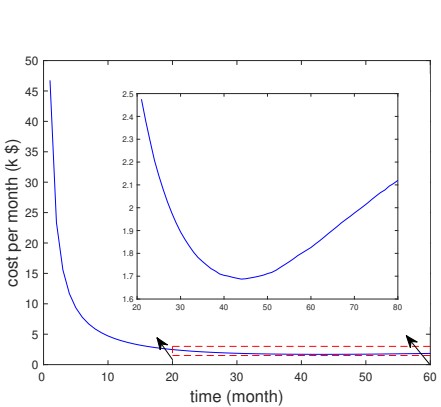

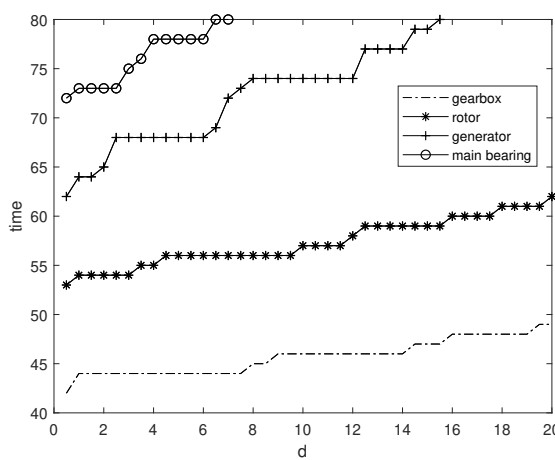

**Figure 1.** Left panel: monthly maintenance cost for the gearbox with the set-up cost $d = 10$. Right panel: $\tau$ as a function of the set-up cost $d$ for different single-component systems.

The left panel of Figure 1 presents a typical profile for the function $a_t$, the inset in the left panel shows that the best time for next PM is at $\tau = 43$ when the set-up cost $d = 5$. The maintenance cost in this case is $a_{43} = 1.7$ (1000 $) per month.

220 Among other things, the graph explains why the choice of $r = 80$ is justified. The right panel of Figure 1 compares the $\tau$ values



obtained for four single-component systems. When the set-up cost $d$ is different, the optimal time to perform the next PM for different components (which are shown in the y-axis, the unit is months) are changed correspondingly.

## 4.2 Study 2: seasonal effects

Part A. To address the seasonal effects of the set-up costs $d_t$, the following set-up costs (in thousands of USD) for different
225 months in a year are used:

| Jan | Feb | Mar | Apr | May | Jun | Jul | Aug | Sep | Oct | Nov | Dec |
|-----|-----|-----|-----|-----|-----|-----|-----|-----|-----|-----|-----|
| 7.5 | 6.5 | 5.5 | 4.5 | 3.5 | 2.5 | 2.5 | 3.5 | 4.5 | 5.5 | 6.5 | 7.5 |

The set-up costs for different months are different, but for the same month at different years are the same. The average set-up cost is $\bar{d} = 5$. Table 2 summarises the results produced by the NextPM algorithm applied to the following three settings:

the winter start scenario (i.e. the wind turbine start function at Jan) with

$$d_1 = 7.5, d_2 = 6.5, \ldots, d_{12} = 7.5, d_{13} = 7.5, d_{14} = 6.5, \ldots,$$

the summer start scenario (i.e. the wind turbine start function at Jul) with

$$d_1 = 2.5, d_2 = 3.5, \ldots, d_{12} = 2.5, d_{13} = 2.5, d_{14} = 3.5, \ldots,$$

the constant set-up cost scenario with $d_1 = 5, d_2 = 5, d_3 = 5, \ldots$

| Component $j$ | 1 | 2 | 3 | 4 | Corresponding month | Monthly maintenance cost |
|---------------|---|---|---|---|---------------------|--------------------------|
| Winter start | x | x | 43 | x | Jul | 4.876 |
| Summer start | 48 | x | 48 | x | Jul | 4.863 |
| Constant set-up cost | 50 | 50 | 50 | 50 | | 4.964 |

**Table 2.** Summary of the NextPM results for $\bar{d} = 5$.

230    According to Table 2, in the Winter start setting, the optimal next PM plan suggests a PM activity on month 43 only for the component 3, the gearbox. With the seasonal set-up cost, the next PM is always planned during the summer since the set-up cost is low then. The most economic among the three scenarios is to start in the summer time, with the optimal plan being to perform the next PM activity on month 48 by replacing the components 1 and 3.

Part B. In this section, the set-up costs are doubled to contrast the results of Part A, so that $\bar{d} = 10$ and $d_t$ takie the following
235 values depending on which month of the year lies behind the time parameter $t$:

| Jan | Feb | Mar | Apr | May | Jun | Jul | Aug | Sep | Oct | Nov | Dec |
|-----|-----|-----|-----|-----|-----|-----|-----|-----|-----|-----|-----|
| 15 | 13 | 11 | 9 | 7 | 5 | 5 | 7 | 9 | 11 | 13 | 15 |





| Component $j$ | 1 | 2 | 3 | 4 | Corresponding month | Monthly maintenance cost |
|---|---|---|---|---|---|---|
| Winter start | 54 | 54 | 54 | 54 | Jun | 5.010 |
| Summer start | 49 | 49 | 49 | 49 | Jul | 4.979 |
| Constant set-up cost | 52 | 52 | 52 | 52 | | 5.061 |

**Table 3.** Summary of the NextPM results for $\bar{d} = 10$.

The new results presented in Table 3 are drastically different from the results of Part A. They suggest (as a consequence of higher set-up costs) to perform PM to all four components at a certain time, irrespective of the scenario. Again, with the summer start setting, the average monthly maintenance cost is somewhat lower. Notice that in all of the seasonal settings, the proposed PM activities are scheduled for summer months (having lower set-up costs).

Part C. A simple wind turbine maintenance strategy is to ignore the PM option and perform a CM activity whenever a turbine component breaks down. This leads to the question: how much can one save by introducing PM planning? The total cost associated with the pure CM strategy is estimated based on the random number of failures over the time interval $[0, T]$ for all $n$ components

$$F(T) = \sum_{j=1}^{n} \mathrm{E}\left( \sum_{i=1}^{\infty} 1_{\{V_i^j \leq T\}} (d_{V_i^j} + b_j) \right) = \sum_{j=1}^{n} \int_0^T (d_u + b_j) dH_j(u), \tag{31}$$

where $H_j$ are the corresponding renewal functions

$$H_j(t) = \mathrm{E}\left( \sum_{i=1}^{\infty} 1_{\{V_i^j \leq t\}} \right), \quad t > 0, \; j = 1, \ldots, n. \tag{32}$$

According to the standard renewal theory, see for example Grimmett et al. (2020), for the large values of $T$,

$$\frac{F(T)}{T} \approx \sum_{j=1}^{n} \frac{1}{T\mu_j} \int_0^T (d_u + b_j) du = \sum_{j=1}^{n} \frac{\bar{d} + b_j}{\mu_j}, \tag{33}$$

where $\bar{d} = \frac{d_1 + \ldots + d_T}{T}$. Applying this approximation to the four-component model of the wind turbine, the monthly maintenance costs for the pure CM strategy are computed to be 7.396 for $\bar{d} = 5$, and 7.618 for $\bar{d} = 10$. Comparison of the costs produced by the NextPM algoritms in Parts A Table 2 and part B Table 3, shows that implementation of the PM planning results in 35% cost saving.

### 4.3 A performance comparison with PMSPIC

Comparison of the NextPM model with the PMSPIC model is not a straightforward exercise since the latter produces a maintenance plan for the whole lifespan $[0, T]$ of the multi-component system in question. The following three tables summarise the results for three values of the constant set-up cost $d$:



| $d = 1$ | 1 | 2 | 3 | 4 | Monthly maintenance cost | Matlab | AMPL |
|---|---|---|---|---|---|---|---|
| NextPM | x | x | 43 | x | 4.731 | 49 sec | 0.01 sec |
| PMSPIC | x | x | 41 | x | 4.749 | 100 sec | 2.25 sec |

| $d = 5$ | 1 | 2 | 3 | 4 | Monthly maintenance cost | Matlab | AMPL |
|---|---|---|---|---|---|---|---|
| NextPM | 50 | 50 | 50 | 50 | 4.964 | 54 sec | 0.01 sec |
| PMSPIC | 51 | 51 | 51 | 51 | 4.884 | 102 sec | 10.62 sec |

| $d = 10$ | 1 | 2 | 3 | 4 | Monthly maintenance cost | Matlab | AMPL |
|---|---|---|---|---|---|---|---|
| NextPM | 52 | 52 | 52 | 52 | 5.061 | 55 sec | 0.01 sec |
| PMSPIC | 47 | 47 | 47 | 47 | 5.025 | 101 sec | 13.47 sec |

The main difference between NextPM and PMSPIC lies in the effectiveness of the algorithms reported in the rightmost columns. For example, if $d = 10$, then the NextPM optimisation runs 10000 times faster than the PMSPIC optimisation.

For $d = 5$, the NextPM calculations are performed with the time unit being three days. The results are rather similar to those obtained for the time unit 1 month. Solving this problem with AMPL has required time increase from 0.01 to 0.06 seconds caused by a ten-fold increase of the number of the time steps. The corresponding increase in the AMPL time for the PMSPIC model was much more dramatic: it takes more than 5 hours to solve it.

## 5 Conclusions

This article introduces a new NextPM optimisation model, which is tested with three case studies based on the data taken from Tian et al. (2011). Under the seasonal variation, the results show that PM activities should be always scheduled in the summer time. This is due to the lower set-up costs during the summer months. When the NextPM model is compared to the pure CM strategy, it is found that around $35\%$ of the maintenance costs can be saved by applying the NextPM model.

In the third case study the NextPM model is compared with the model PMSPIC from Gustavsson et al. (2014).The comparison demonstrated that the NextPM model is accurate, profitable, and much less complex than the PMSPIC, enabling to use a much shorter time interval.

Since the NextPM model is both accurate and fast to solve, the algorithm stemming from the proposed model can be used as a key module in a maintenance scheduling app.



## Appendix A:  Complete optimisation model of NextPM

$$\text{minimize } f(\boldsymbol{z}, \boldsymbol{x}^1, \ldots, \boldsymbol{x}^n) := \sum_{t=s+1}^{r+1} \frac{1}{t-s} \Big( d_t z_t + c_{s,t}^1 x_t^1 + \ldots + c_{s,t}^n x_t^n \Big),$$

subject to

$$\sum_{t=s+1}^{r+1} x_t^j = 1, \qquad j = 1, \ldots, n,$$

$$z_t \geq x_t^j, \qquad t = s+1, \ldots, r+1, \ j = 1, \ldots, n,$$

$$D_{s,t}^j x_t^j \geq 0, \qquad t = s+1, \ldots, r, \ j = 1, \ldots, n,$$

$$z_t \in \{0,1\}, \quad t = s+1, \ldots, r+1,$$

$$x_t^j \in \{0,1\}, \quad t = s+1, \ldots, r+1, \ j = 1, \ldots, n.$$

## Appendix B:  Complete optimisation model of NextOM

$$\text{minimize } f(\boldsymbol{z}, \boldsymbol{x}^1, \ldots, \boldsymbol{x}^n) := \sum_{t=s+1}^{s+2} \frac{1}{t-s} \Big( d_t z_t + \sum_{j \neq i} c_{s,t}^j x_t^j \Big),$$

subject to

$$\sum_{t=s+1}^{s+2} x_t^j = 1, \qquad j = 1 \ldots, n,$$

$$D_{s,s+1}^j x_{s+1}^j \geq 0, \qquad j = 1, \ldots, i-1, i+1, \ldots, n,$$

$$x_{s+1}^{(i)} = 1,$$

$$z_t \geq x_t^j, \qquad t = s+1, s+2, \ j = 1, \ldots, n,$$

$$z_t \in \{0,1\}, \quad t = s+1, s+2,$$

$$x_t^j \in \{0,1\}, \quad t = s+1, s+2, \ j = 1, \ldots, n.$$

*Author contributions.*  Quanjiang Yu developed the theoretical formalism, performed the analytic calculations and performed the numerical simulations. Both Quanjiang Yu , Michael Patriksson , and Serik Sagitov contributed to the final version of the manuscript.

*Competing interests.*  There is no significant competing financial, professional, or personal interests that might have influenced the performance or presentation of the work described in this manuscript.



*Acknowledgements.* We acknowledge the financial support from the Swedish Wind Power Technology Centre at Chalmers, and from the Swedish Research Council (Dnr. 2014-5138). Special thanks to the director of SWPTC, professor Ola Carlson, for his constructive recommendations that considerably improved the final layout of the paper.





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
