# Peer review of "Optimal scheduling of the next preventive maintenance activity for a wind farm"

_Wind Energy Science, 2020_

## Referee Comment (RC1) · Miriam Noonan (Referee) · 4 Jan 2021

This was an interesting paper on a topic of great interest to the industry right now looking at extending the life of assets whilst minimising operational costs. A few small notes I picked up during the review:

Line 13 – Define CM as corrective maintenance Line 41 – Would 'mobilization costs' work better than 'set-up costs'? Line 88 – OM isn't clearly defined in the paper – might be worth clearly stating as 'operational maintenance' Line 203 – May be worth stating this relates to onshore wind farms – 20 years seems low for an offshore project.

Kind Regards, Miriam Noonan

---

## Author Comment (AC1) · 4 Jan 2021

Dear Professor Miriam Noonan,

Thank you for your valuable comments. Here I reply to your comments one by one.

Comment 1: Line 13 – Define CM as corrective maintenance Response 1: Yes, I am changing 'Among other things, this analysis reveals a dramatic cost reduction achieved by the NextPM model as compared to the the pure CM strategy.' to 'Among other things, this analysis reveals a dramatic cost reduction achieved by the NextPM model as compared to the the pure corrective maintenance strategy.'

Comment 2: Line 41 – Would 'mobilization costs' work better than 'set-up costs'? Response 2: Yes, I am going to change 'set-up costs' to 'mobilization costs'.

Comment 3: Line 88 – OM isn't clearly defined in the paper – might be worth clearly stating as 'operational maintenance' Response 3: On line 85: 'the possibility of a component failure before the planned PM, followed by opportunistic maintenance (OM) activities.' OM is defined as opportunistic maintenance. Hopefully this answers Comment 3.

Comment 4: Line 203 – May be worth stating this relates to onshore wind farms – 20 years seems low for an offshore project. Response 4: Yes, I am changing 'The lifetime of the wind turbine is assumed to be 20 years, which is the typical case in the industry now, according to Ziegler et al. (2018).' to 'The lifetime of the wind turbine is assumed to be 20 years, which is a typical life length for onshore wind farms, according to Ziegler et al. (2018).'

Best regards Quanjiang Yu

---

## Referee Comment (RC2) · Jonas Kaczenski (Referee) · 13 Jan 2021

In the O&M phase, the operating and maintenance costs for the planners and operators of offshore wind farms are associated with a high-risk potential and uncertainties. Interesting approach to optimize the scheduling of preventive maintenance and reduce operational costs.

You will find just a few comments below:

- Mobilization costs could be a better fit than set-up costs - Define CM as corrective maintenance in line 13 instead of line 27 - Define OM as opportunistic maintenance in line 38 instead of line 85 - Maybe state that Ziegler et al. (2018) is about Lifetime extension of onshore wind turbines - I would prefer an equally sized left panel in comparison to the right panel - Could you refer to a source for the mobilization costs in line 226?

Congratulations and best regards, Jonas Kaczenski

---

## Author Comment (AC2) · 16 Jan 2021

Comment 1: Mobilization costs could be a better fit than set-up costs Response 1: Yes, I have already changed 'set-up costs' to 'mobilization costs'.

Comment 2: Define CM as corrective maintenance in line 13 instead of line 27 Response 2: Yes, I am changing 'Among other things, this analysis reveals a dramatic cost reduction achieved by the NextPM model as compared to the pure CM strategy. ' to 'Among other things, this analysis reveals a dramatic cost reduction achieved by the NextPM model as compared to the pure corrective maintenance (CM) strategy. '

Comment 3: Define OM as opportunistic maintenance in line 38 instead of line 85 Response 3: Yes, I am changing 'The article Sarker and Faiz (2016) looks at opportunistic

maintenance which is a special kind of preventive maintenance.' to 'The article Sarker and Faiz (2016) looks at opportunistic maintenance (OM) which is a special kind of preventive maintenance.'

Comment 4: Maybe state that Ziegler et al. (2018) is about Lifetime extension of onshore wind turbines Response 4: Yes, I am changing 'The lifetime of the wind turbine is assumed to be 20 years, which is the typical case in the industry now, according to Ziegler et al. (2018).' to 'The lifetime of the wind turbine is assumed to be 20 years, which is a typical life length for onshore wind farms, according to Ziegler et al. (2018).'

Comment 5: I would prefer an equally sized left panel in comparison to the right panel Response 5: Yes, now they are equally sized.

Comment 6: Could you refer to a source for the mobilization costs in line 226? Response 6: I have added a source for the mobilization costs. Attached is the updated version of case study 2.

Please also note the supplement to this comment:
https://wes.copernicus.org/preprints/wes-2020-129/wes-2020-129-AC2-supplement.pdf

**Supplement:**

**0.1 Study 2: seasonal effects**

Part A. To address the seasonal effects of the mobilization costs $d_t$, the following mobilization costs (in thousands of USD) for different months in a year are used:

| Jan | Feb | Mar | Apr | May | Jun | Jul | Aug | Sep | Oct | Nov | Dec |
|-----|-----|-----|-----|-----|-----|-----|-----|-----|-----|-----|-----|
| 15 | 13 | 11 | 9 | 7 | 5 | 5 | 7 | 9 | 11 | 13 | 15 |

The mobilization costs for different months are different, but for the same month at different years are the same. The average mobilization cost is $\bar{d} = 10$. These numbers are obtained based on a discussion with the experts within the Swedish Wind Power Technology Centre (SWPTC). Table 1 summarises the results produced by the NextPM algorithm applied to the following three settings:

the winter start scenario (i.e. the wind turbine start function at Jan) with

$$d_1 = 7.5, d_2 = 6.5, \ldots, d_{12} = 7.5, d_{13} = 7.5, d_{14} = 6.5, \ldots,$$

the summer start scenario (i.e. the wind turbine start function at Jul) with

$$d_1 = 2.5, d_2 = 3.5, \ldots, d_{12} = 2.5, d_{13} = 2.5, d_{14} = 3.5, \ldots,$$

the constant mobilization cost scenario with $d_1 = 5, d_2 = 5, d_3 = 5, \ldots$

| Component $j$ | 1 | 2 | 3 | 4 | Corresponding month | Monthly maintenance cost |
|---------------|----|----|----|----|---------------------|--------------------------|
| Winter start | 54 | 54 | 54 | 54 | Jun | 5.010 |
| Summer start | 49 | 49 | 49 | 49 | Jul | 4.979 |
| Constant mobilization cost | 52 | 52 | 52 | 52 | – | 5.061 |

**Table 1.** Summary of the NextPM results for $\bar{d} = 10$.

They suggest (as a consequence of high mobilization costs) to perform PM to all four components at a certain time, irrespective of the scenario. With the summer start setting, the average monthly maintenance cost is somewhat lower. Notice that in all of the seasonal settings, the proposed PM activities are scheduled for summer months (having lower mobilization costs).

Part B. In this section, the mobilization costs are halved to contrast the results of Part A, so that $\bar{d} = 5$ and $d_t$ take the following values depending on which month of the year lies behind the time parameter $t$:

| Jan | Feb | Mar | Apr | May | Jun | Jul | Aug | Sep | Oct | Nov | Dec |
|-----|-----|-----|-----|-----|-----|-----|-----|-----|-----|-----|-----|
| 7.5 | 6.5 | 5.5 | 4.5 | 3.5 | 2.5 | 2.5 | 3.5 | 4.5 | 5.5 | 6.5 | 7.5 |

The new results presented in Table 2 are drastically different from the results of Part A.

| Component $j$ | 1 | 2 | 3 | 4 | Corresponding month | Monthly maintenance cost |
|---------------|----|----|----|----|---------------------|--------------------------|
| Winter start | x | x | 43 | x | Jul | 4.876 |
| Summer start | 48 | x | 48 | x | Jul | 4.863 |
| Constant mobilization cost | 50 | 50 | 50 | 50 | – | 4.964 |

**Table 2.** Summary of the NextPM results for $\bar{d} = 5$.

According to Table 2, in the Winter start setting, the optimal next PM plan suggests a PM activity on month 43 only for the component 3, the gearbox. With the seasonal mobilization cost, the next PM is always planned during the summer since the

mobilization cost is low then. Again, the most economic among the three scenarios is to start in the summer time, with the optimal plan being to perform the next PM activity on month 48 by replacing the components 1 and 3.

Part C. A simple wind turbine maintenance strategy is to ignore the PM option and perform a CM activity whenever a turbine component breaks down. This leads to the question: how much can one save by introducing PM planning? The total cost associated with the pure CM strategy is estimated based on the random number of failures over the time interval $[0, T]$ for all $n$ components

$$F(T) = \sum_{j=1}^{n} \mathrm{E}\left( \sum_{i=1}^{\infty} 1_{\{V_i^j \leq T\}}(d_{V_i^j} + b_j) \right) = \sum_{j=1}^{n} \int_0^T (d_u + b_j) dH_j(u), \tag{1}$$

where $H_j$ are the corresponding renewal functions

$$H_j(t) = \mathrm{E}\left( \sum_{i=1}^{\infty} 1_{\{V_i^j \leq t\}} \right), \quad t > 0, \; j = 1, \ldots, n. \tag{2}$$

According to the standard renewal theory, see for example Grimmett et al. (2020), for the large values of $T$,

$$\frac{F(T)}{T} \approx \sum_{j=1}^{n} \frac{1}{T \mu_j} \int_0^T (d_u + b_j) du = \sum_{j=1}^{n} \frac{\bar{d} + b_j}{\mu_j}, \tag{3}$$

where $\bar{d} = \frac{d_1 + \ldots + d_T}{T}$. Applying this approximation to the four-component model of the wind turbine, the monthly maintenance costs for the pure CM strategy are computed to be 7.396 for $\bar{d} = 5$, and 7.618 for $\bar{d} = 10$. Comparison of the costs produced by the NextPM algoritms in Parts A Table 2 and part B Table 1, shows that implementation of the PM planning results in 35% cost saving.

**References**

35   Grimmett, G. S. et al.: Probability and random processes, Oxford university press, 2020.

---

## Editor Comment (EC1) · Katherine Dykes (Editor) · 28 Jan 2021

This comment from an anonymous reviewer came after the comment period closed:

Review for manuscript: wes-2020-129 "Optimal scheduling of the next preventive maintenance activity for a wind farm" This paper considers the interesting problem of maintenance for wind turbines. Effectively managing turbine maintenance is a key problem for wind farm operators, and has the potential impact of substantially lowering metrics such as the LCOE for wind energy. The authors present an optimisation model for the problem of maintaining a single turbine. They then present three examples of applying the models with different parameterisations. Overall, I am unclear on what contribution this paper makes to the literature on maintenance in general, or turbine

maintenance in particular. The problem context that is presented – maintenance of a single turbine comprising four components – seems like a substantial simplification of the actual problem facing wind farm operators, and so some reflection on how this work would actually be applied in a practical setting would be useful. Section 1: in the discussion on maintenance literature, it will be useful to be clearer which papers are specific to wind turbine maintenance, as it currently reads as if these are all generic maintenance studies. More broadly, it would be useful to have more of a review of the literature on turbine maintenance – there is a substantial body of research in this area, and it is unclear exactly what the contribution of this paper is. Section 3.5: the differences between the authors work and the PMSPIC model could be clearer – this section presents the mathematical formulation PMSPIC, but more reflection on how these differences can be interpreted would be useful, including the differences which give rise to the substantial differences in computing times that are mentioned in Section 4. Section 4: It would be useful to clarify whether the data used in the numerical studies is intended to represent onshore or offshore turbine maintenance (there is no clear statement in the paper as to which type of problem this model is intended for). The statement " Among other things, the graph explains why the choice of r = 80 is justified" should be clarified. I'm not clear as to the purpose of the three studies, or the sub-parts within the studies. It would be useful to have the analysis approach set out more clearly to emphasise what the authors are aiming to demonstrate. Tables 4-6 aren't numbered and don't have any captions. Its not clear to me how the discussion and interpretation in Section 4.3 relate to the data that us shown in the tables. There is no real explanation on what the "matlab" and "AMPL" times actually represent. Also, the PMSPIC algorithm seems to identify solutions which are more optimal, but there is no reflection on this. If I am correctly understanding the optimal PM times that are shown throughout Section 4, these seem extremely long (e.g. 4 years plus). Typical planned maintenance schedules for turbines are much more frequent, so it would be useful to have some reflection on why these results are so different.

---

## Author Response (AR1)

Comment 1: Line 13 – Define CM as corrective maintenance

Response 1: I changed 'Among other things, this analysis reveals a dramatic cost reduction achieved by the NextPM model as compared to the the pure CM strategy.' to 'Among other things, this analysis reveals a dramatic cost reduction achieved by the NextPM model as compared to the the pure corrective maintenance (CM) strategy.'

Comment 2: Line 41 – Would 'mobilization costs' work better than 'set-up costs'?

Response 2: I changed 'set-up costs' to 'mobilization costs'.

Comment 3: Define OM as opportunistic maintenance in line 38 instead of line 85

Response 3: I changed 'The article Sarker and Faiz (2016) looks at opportunistic maintenance which is a special kind of preventive maintenance.' to 'The article Sarker and Faiz (2016) looks at opportunistic maintenance (OM) which is a special kind of preventive maintenance.'

Comment 4: Line 203 – May be worth stating this relates to onshore wind farms – 20 years seems low for an offshore project.

Response 4: I changed 'The lifetime of the wind turbine is assumed to be 20 years, which is the typical case in the industry now, according to Ziegler et al. (2018).' to 'The lifetime of the wind turbine is assumed to be 20 years, which is a typical life length for onshore wind farms, according to Ziegler et al. (2018).'

Comment 5: I would prefer an equally sized left panel in comparison to the right panel

Response 5: Now they are equally sized.

Comment 6: Could you refer to a source for the mobilization costs in line 226?

Response 6: I have added a source for the mobilization costs.

---

## Editor Decision (ED1)

Wes-2020-129

Overall
- The article is very abstract and very much written as if intended for an OR audience. WES is an application-oriented journal where the domain focus of wind energy is the common denominator – thus, papers should be rooted in the realism of the domain. This should be improved substantially in the paper and several suggestions to do so are made by section.
- The introduction does not clearly state what NextPM is and how it extends the state of the art. This should be crystal clear. Flow diagrams may help. Or using a motivating real example and then comparing for that example how NextPM will work as contrasted to PMSPIC.
- The lack of wind-specific context in the entire methods portion of the paper (sections 2 and 3) is a problem – see detailed notes below
- Section 4 also is quite abstracted from reality and some of the data uses is old or model parameterization not thoroughly justified. Different turbine components are listed but there is no discussion of them (figure 1)
- Have a native English speaker edit the paper for syntax – a lot is well written but there are also many places where sentences are hard to read and I had to read them multiple times to get their intent.

Abstract
- Discussion on global warming in abstract is unnecessary – okay for introduction but abstract should be succinct and to the point, suggest eliminating entire first paragraph
- The abstract reads too much like a marketing pitch for NextPM – I suggest rewording it to put more emphasis on the scientific contribution to the state of the art

Introduction
- Lazard 2017 is a bit outdated – suggest updating reference to most recent Lazard.
- Citation on O&M costs for offshore as percent of total? There are many – see cost of wind energy review from NREL or various other sources
- The literature review is okay but there is a decent amount of literature in this space that is not addressed. Suggest the author look at the work from U. Strathclyde, ECN/TNO O&M calculator, Fraunhofer IWES, work by Dimitrov et al, and commercial capability of Peak Wind/Lautec/SeaImpact to supplement existing literature base
    o Particularly the O&M calculator from TNO provides a breakdown of planned and unplanned maintenance as well as a breakdown of failure types with a range of severity from minor repair to component replacement. It seems in the paper you are focusing on component replacement – yes? Be a bit more clear about this
    o Most maintenance models like the ones from the organizations above are time-series based which makes optimization of scheduling quite difficult – though there are efforts out there even in commercial tools
- Define what you mean by multi-component systems – it is somewhat obvious but important

- How are the models in Moghaddam and Usher nonlinear?
- It is not clear what the PMSPIC model does at all – flow diagrams would be VERY helpful to illustrate what that model does and what your model does (in order to contrast the two effectively) – already in the introduction you should be stating at a high level how the NextPM algorithm is different from and improves upon PMSPIC
- It seems like the lines 53 to 57 are a bit thrown in there.
- It would be helpful to have a bit more of the big picture to better contextualize the work. Here is a suggested path for reworking the introduction:
  - Start with the real-world issue: offshore wind O&M is costly – how costly? See NREL work for example – therefore want to reduce
  - How does O&M for offshore wind work? again, see ECN O&M calculator as a good high level overview of how offshore wind O&M works in practice
  - But, many of these models rely on time-series simulation of the failures and logistics where there are many underlying probabilistic / uncertain events both with component failures (of various types) and access issues (related to weather conditions)
  - Thus, this becomes a very challenging optimization under uncertainty problem
  - How have people tackled this before?
    - In general, operations research communities have looked at O&M optimization via several techniques
    - For wind specifically, optimization of O&M has been tackled in x,y, z ways
  - In this paper, we build on the state of the art with a new algorithm, NextPM, that does x,y,z and thus has potential for capturing a,b,c, additional realism of the offshore wind O&M optimization problem and/or improves the computational efficiency by leveraging d,e,f…
  - The paper is organized as follows…
- Appendix A and B have only the formal optimization problem formulations – is it necessary to have them as two separate appendices? Or appendices at all? Personally I prefer to see the problem formulation in situ and up front as it is the central organizing basis for an optimization study

Optimal rescheduling algorithm
- Before jumping to the description of the algorithm, consider describing it at a high-level in plan language
- FYI: A key cost of offshore wind failures is not the component costs but the downtime and associated loss of energy production. Most of the O&M models mentioned above also calculate availability which can then translate directly to revenue losses
- FYI: most models above also treat dt is a uncertain variable since it is highly weather dependent, bj and cj are also uncertain but models often treat them as deterministic – i.e. failures are probabilistic but the costs to repair them are deterministic
- Line 85 – need to explain here in plain terms what lamda is
- Again, I think a flow diagram would help here. Can you represent visually or in plan language what the optimization problem you are trying to solve is? Contextualize it

with an example using a wind turbine? And contrast it with a baseline approach? It is still unclear what is novel here.
-   I am not sure why section 2 is a stand-alone section. It seems to me it is part of the methods and I recommend section 2 and 3 be merged and made subsections of an overarching methods section
-   It is still not clear what NextPM is doing by the end of section 2 nor NextOM

Optimal plan for next preventive maintenance
-   Overall section 3 would benefit from more contextualization from the real world offshore wind O&M problem. Nothing about section 3 seems tied specifically to the offshore wind O&M optimization problem. – can you bring in real world examples to help contextualize the approach and demonstrate its uniqueness and value from an offshore wind problem-specific perspective?
-   Line 101-103 – this is important and should be discussed in the introduction. Also, it should be explained WHY this approach is being taken. It is not self-evident.
-   Line 104: Explain what z and x are from a real-world perspective
-   Line 110: why? Explain why this approach is being taken
-   Line 142 and 152 – why is lambda introduced? Please explain more clearly
-   Lines 212-215 should be brought into the introduction. It should already be clear up front how this work will extend the state of the art (though details can remain in a later section)
-   Also, line 212 is the FIRST mention in all of section 2 and 3 of a wind turbine. Nothing about the work to this point is tied to the offshore wind problem specific context… this is a problem for publishing in WES and needs to be remedied before acceptance to publication (see earlier notes on same topic)

Numerical studies
-   There are much more recent papers related to wind turbine failures rates and repair costs – see work from organizations mentioned above and also reporting from NREL, ORE Catapult, BVG Associates, and others
-   Typical farm lifes are more like 25 to 30 or longer – lifetime from a financing perspective is often 20 but has been getting longer
-   Line 224 – explain this lambda value better– at least in a footnote if not in a paragraph. What computer simulations elsewhere and why not reported???
-   Line 223 and line 235 – the decision of the specification of r is not well explained
-   Line 239 – d= 5, 5 what?
-   Lines 240-245, for wind systems, components are designed for the full plant lifetime (in this case 20 years). Any component replacement prior to that is considered a premature failure and thus is typically in the bucket of unplanned maintenance. Condition monitoring can help detect components that will fail prematurely, which is where the approach in this paper would become relevant. But it is odd to say that PM replacement (for any major component is 43 months) this would be SUPER short from an industry perspective. Again, this entire work seems pretty unlinked to the wind specific O&M problem
-   Figure 1 shows the components but neither of the two figures are really well explained. It would be good to separate these into 2 figures and clearly explain what each of them means

- Line 250- Mobilization cost of d = 10, 10 what?
- Part A and B and C headers (line 252, line 264 and line 272 respectively), make these separate lines and not in paragraph text
- Explain better what you mean by winter versus summer start – does this make sense from a real world problem perspective?
- I recommend doing Part C first – this would be the baseline (all corrective no predictive maintenance) – part A introduces some level of PM and then part B considers the seasonal effects. Then, include the percent savings in each of table 2 and 3 and reference back to the CM analysis
- Need to explain lines 286 to 289 better. How can you do a fair comparison?
- Line 304 – 5 hours to solve what? The full optimization problem?
- The overall comparison of NextPM and PMSPIC seems incomplete – are there caveats to this? What are the key assumptions you are making that might be limiting the external validity of the work?

Conclusions
- Consider revisiting the conclusions after the rest of the paper is reworked and better tied to the wind energy problem
- Do not mention preliminary results (not shown) on computational time… either show them or don't mention them – i.e. leave it as future work
- Also, what is the future work? where will this effort go? What will it take to make this actually useful for wind farm O&M planning?
- Line 316-317 again is more like a marketing pitch

---

## Author Response (AR2)

**Reply to the comments of the reviewers**

We are thankful to the reviewers for the close reading of the manuscript and the valuable comments. Below comes the full list of reviewers' comments and the changes we made in response.

**Comment 1.** Line 13 – Define CM as corrective maintenance

Change 1: at line 13, we changed

"Among other things, this analysis reveals a dramatic cost reduction achieved by the NextPM model as compared to the the pure CM strategy."

to

"Among other things, this analysis reveals a dramatic cost reduction achieved by the NextPM model as compared to the the pure corrective maintenance (CM) strategy."

**Comment 2.** Line 41 – Would 'mobilization costs' work better than 'set-up costs'?

Change 2: we changed "set-up costs" to "mobilization costs" everywhere.

**Comment 3.** Define OM as opportunistic maintenance in line 38 instead of line 85

Change 3: at line 38, we changed

"The article Sarker and Faiz (2016) looks at opportunistic maintenance which is a special kind of preventive maintenance."

to

"The article Sarker and Faiz (2016) looks at opportunistic maintenance (OM) which is a special kind of preventive maintenance."

**Comment 4.** Line 203 – May be worth stating this relates to onshore wind farms – 20 years seems low for an offshore project.

Change 4: at line 203, we changed

"The lifetime of the wind turbine is assumed to be 20 years, which is the typical case in the industry now, according to Ziegler et al. (2018)."

to

"The lifetime of the wind turbine is assumed to be 20 years, which is a typical life length for onshore wind farms, according to Ziegler et al. (2018)."

**Comment 5.** I would prefer an equally sized left panel in comparison to the right panel

Change 5: we changed the size of the figure such that they are equally sized now.

**Comment 6.** Could you refer to a source for the mobilization costs in line 226?

Change 6: at line 239, we added

"These numbers are obtained based on a discussion with the experts within the Swedish Wind Power Technology Centre (SWPTC)."

**Comment 7.** Overall, I am unclear on what contribution this paper makes to the literature on maintenance in general, or turbine maintenance in particular. The problem context that is presented – maintenance of a single turbine comprising four components – seems like a substantial simplification of the actual problem facing wind farm operators, and so some reflection on how this work would actually be applied in a practical setting would be useful.

Change 7: after the line 48, we add

"However, the PMSPIC model is very complex and solving it takes a long time. This motivated us to build a new optimisation model which would be both accurate and could be solved really fast. For simplicity, our modelling idea is presented in the framework of a single turbine as multiple component system, however this framework with a little effort can be extended to the setting of several wind turbine farms."

Change 8: after line 278, we add a new paragraph

"In this paper our NextPM model is applied to a system of four components belonging to a single wind turbine. However, we claim that our approach can handle the case of, say, ten turbines with 80 components in total. Preliminary results (not shown) demonstrate that the computational time required by our algorithm grows linearly with the increased number of components, while the PMSPIC's computational time grows exponentially fast."

**Comment 8**. Section 1: in the discussion on maintenance literature, it will be useful to be clearer which papers are specific to wind turbine maintenance, as it currently reads as if these are all generic maintenance studies. More broadly, it would be useful to have more of a review of the literature on turbine maintenance – there is a substantial body of research in this area, and it is unclear exactly what the contribution of this paper is.

Change 9. We add four papers to the list of references:

[1] Zheng R, Zhou Y, Zhang Y. Optimal preventive maintenance for wind turbines considering the effects of wind speed[J]. Wind Energy, 2020, 23(11): 1987-2003.
[2] Davoodi A, Peyghami S, Yang Y, et al. A Preventive Maintenance Planning Approach for Wind Converters[C]//2020 5th IEEE Workshop on the Electronic Grid (eGRID). IEEE, 2020: 1-8.
[3] Wang J, Zhang X, Zeng J, et al. Optimal dynamic imperfect preventive maintenance of wind turbines based on general renewal processes[J]. International Journal of Production Research, 2020, 58(22): 6791-6810.
[4] Zhang C, Gao W, Guo S, et al. Opportunistic maintenance for wind turbines considering imperfect, reliability-based maintenance[J]. Renewable energy, 2017, 103: 606-612.

Change 10. Before line 49 we add

"The recent literature on wind turbine preventive maintenance planning extends the modelling scope by paying special attention to particular performance factors for the wind power systems. Zheng et al. (2020) look into the effects of the varying wind speed on the wind turbine maintenance planning. Davoodi et al. (2020) single out the converter as a crucial component of the wind turbine and builds an optimization model to find the optimal replacement times for the converters. Wang et al. (2020) and Zhang et al. (2017) deal with imperfect preventive maintenance. Meanwhile, the main concern of our paper is the computational time of the optimization model. An optimization algorithm with drastically reduced computational time can be used as a key module in a maintenance scheduling app."

**Comment 9.** Section 3.5: the differences between the authors work and the PMSPIC model could be clearer – this section presents the mathematical formulation PMSPIC, but more reflection on how these differences can be interpreted would be useful, including the differences which give rise to the substantial differences in computing times that are mentioned in Section 4

Change 11. I added a new paragraph after line 200

"The main difference between PMSPIC and the NextPM model is that the PMSPIC generates a maintenance plan for the whole lifetime of the wind turbine, while the NextPM model produces an optimal schedule only for the next PM activity. By focusing on one PM activity at a time and implementing a different model structure we succeeded in substantial reduction of the computational time."

**Comment 10.** The statement "Among other things, the graph explains why the choice of r = 80 is justified" should be clarified.

Change 12. The line 207

"r = 80, see Section 4.1 for motivation,"

is replaced by

"r = 60, see Section 4.1 for motivation,"

Change 13. After the line 216 we add

"Here, $a_t$ describes the monthly maintenance cost in the single component setting, if the next PM is planned at time t (assuming that at time 0 the component was as good as new). In this section we analyze the behavior of the function $a_t$ under some realistic model parameters. As a result, we propose r=60 as a practical length of the planning period for our algorithm."

Change 14. Line 221

"Among other things, the graph explains why the choice of r = 80 is justified"

is replaced by

"A smaller value of the parameter r would reduce the computational time of the NextPM model. On the other hand, from the perspective of the Algorithm 1, it is desirable to choose r such that at least one PM activity is scheduled during the planning horizon. Since the observed optimal time to perform the next PM is at month 43, we propose setting r=60."

**Comment 11.** I'm not clear as to the purpose of the three studies, or the sub-parts within the studies. It would be useful to have the analysis approach set out more clearly to emphasise what the authors are aiming to demonstrate.

See Change 13 for Study 1.

Change 15. Concerning Study 2 we added a paragraph after the line 224

"In this section, we study how different mobilization costs $d_t$ result in different optimal PM schedules. Part A deals with the seasonally changing $d_t$ with the average bar d =10. Part B takes up a similar case with a lower bar d = 5. In Part C we compare the next PM plan with a pure CM strategy."

Change 16. Concerning the third case study, after the line 267 we add

"In this case study, we compare the outputs of the NextPM model and the optimization model PMSPIC."

**Comment 12.** Tables 4-6 aren't numbered and don't have any captions.

Change 17. We added numbers and captions.

"Table 4: Outputs of the NextPM and PMSPIC models for d=1."

"Table 5: Outputs of the NextPM and PMSPIC models for d=5."

"Table 6: Outputs of the NextPM and PMSPIC models for d=10."

**Comment 13.** It's not clear to me how the discussion and interpretation in Section 4.3 relate to the data that us shown in the tables. There is no real explanation on what the "matlab" and "AMPL" times actually represent. Also, the PMSPIC algorithm seems to identify solutions which are more optimal, but there is no reflection on this.

Change 18. The paragraph starting at the line 265

"The main difference between NextPM and PMSPIC lies in the effectiveness of the algorithms reported in the rightmost columns. For example, if d= 10, then the NextPM optimisation runs 10000 times faster than the PMSPIC optimisation."

is replaced by

"Tables 4-6 reveal that the next PM schedules produced by NextPM and PMSPIC are quite similar. The observed small differences in the maintenance costs do not imply that PMSPIC gives better solutions, since NextPM calculates the maintenance costs within a different modelling framework.

The main advantage of NextPM compared to PMSPIC is in the computational speed. The effectiveness of the algorithms is reported in the two rightmost columns. The "Matlab" column gives the time it takes to generate the main parameters of the model. For the NextPM the number of parameters is much smaller, and they are $c_{s,t}^j$ $D_{s,t}^j$. The "AMPL" column gives the time it takes to solve the optimisation model. For example, if d= 10, the NextPM optimisation runs 10000 times faster than the PMSPIC optimisation."

**Comment 14.** If I am correctly understanding the optimal PM times that are shown throughout Section 4, these seem extremely long (e.g. 4 years plus). Typical planned maintenance schedules for turbines are much more frequent, so it would be useful to have some reflection on why these results are so different.

Change 19. After line 220, we add

"Notice that by preventive maintenance, we don't mean the practice of regular inspection of the components' condition. Our concern is the optimal planning of preventive replacements of the components based on their age. In this case study the starting age of the component is zero, which partially explains the seemingly long next PM replacement time of 43 months."

---

## Author Response (AR3)

**Reply to the comments of the final reviewer**

We are thankful to the final reviewer for the close reading of the manuscript and the valuable comments. Inspired by reviewer's constructive suggestions, we have made a *major revision of the manuscript* with way too many minor changes in the text to be all shown in this report. This is also the reason we didn't submit a marked-up manuscript version showing the changes made.

Below comes the full list of reviewer's comments with our responses.

**Overall**

**Comment 1**. The article is very abstract and very much written as if intended for an OR audience. WES is an application-oriented journal where the domain focus of wind energy is the common denominator – thus, papers should be rooted in the realism of the domain. This should be improved substantially in the paper and several suggestions to do so are made by section.

**Response 1.** To address this general comment concerning somewhat abstract nature of our paper, we removed some of the technical parts and added three flowcharts. We did our best (under the high time pressure for the corresponding author working on his PhD dissertation and with another coauthor being on a long-term sick leave) to constructively respond to the majority of the more detailed comments and suggestions of the reviewer. We hope that the revised article could be published in your journal.

**Comment 2**. The introduction does not clearly state what NextPM is and how it extends the state of the art. This should be crystal clear. Flow diagrams may help. Or using a motivating real example and then comparing for that example how NextPM will work as contrasted to PMSPIC.

**Response 2.** Many changes in the text address this comment. In particular, we added three flow diagrams.

**Comment 3**. The lack of wind-specific context in the entire methods portion of the paper (sections 2 and 3) is a problem – see detailed notes below

**Response 3.** See the responses to the detailed comments of the reviewer.

**Comment 4**. Section 4 also is quite abstracted from reality and some of the data uses is old or model parameterization not thoroughly justified. Different turbine components are listed but there is no discussion of them (figure 1)

**Response 4.** See Response 1.

**Comment 5**. Have a native English speaker edit the paper for syntax – a lot is well written but there are also many places where sentences are hard to read and I had to read them multiple times to get their intent.

**Response 5.** We did our best to clarify the unclear parts of the paper.

**Abstract**

**Comment 6**. Discussion on global warming in abstract is unnecessary – okay for introduction but abstract should be succinct and to the point, suggest eliminating entire first paragraph.

**Response 6.** The paragraph is removed.

**Comment 7**. The abstract reads too much like a marketing pitch for NextPM – I suggest rewording it to put more emphasis on the scientific contribution to the state of the art.

**Response 7.** The abstract is rewritten.

**Introduction**

**Comment 8.** Lazard 2017 is a bit outdated – suggest updating reference to most recent Lazard.

**Response 8.** The reference is replaced by Lazard 2020.

**Comment 9.** Citation on O&M costs for offshore as percent of total? There are many – see cost of wind energy review from NREL or various other sources

**Response 9.** We have edited:

"A large part of the total cost associated with wind turbines is due to operation and maintenance: 34% for the fixed-bottom offshore wind turbines, according to \cite{stehly20202019}."

**Comment 10.** The literature review is okay but there is a decent amount of literature in this space that is not addressed. Suggest the author look at the work from U. Strathclyde, ECN/TNO O&M calculator, Fraunhofer IWES, work by Dimitrov et al, and commercial capability of Peak Wind/Lautec/SeaImpact to supplement existing literature base.

Particularly the O&M calculator from TNO provides a breakdown of planned and unplanned maintenance as well as a breakdown of failure types with a range of severity from minor repair to component replacement. It seems in the paper you are focusing on component replacement – yes? Be a bit more clear about this.

Most maintenance models like the ones from the organizations above are time-series based which makes optimization of scheduling quite difficult – though there are efforts out there even in commercial tools

**Response 10.** We have added one reference from U. Strathclyde, Browell (2016).

**Comment 11.** Define what you mean by multi-component systems – it is somewhat obvious but important

**Response 11.** We have added

"In the context of wind farm maintenance, each wind turbine is viewed here as a system comprising multiple components such as the gearbox, power generator, rotor and main bearing. Whenever one of the components is broken, the whole system stops functioning. …"

**Comment 12.** How are the models in Moghaddam and Usher nonlinear?

**Response 12.** Both the objective function and the constrains of the optimization models by Moghaddam and Usher are non-linear. A reference to \cite{andreasson2020introduction} is added.

**Comment 13.** It is not clear what the PMSPIC model does at all – flow diagrams would be VERY helpful to illustrate what that model does and what your model does (in order to contrast the two

effectively) – already in the introduction you should be stating at a high level how the NextPM algorithm is different from and improves upon PMSPIC

**Response 13.** We made a significant effort to clarify this at different parts of the article. In particular, see the second flowchart.

**Comment 14.** It seems like the lines 53 to 57 are a bit thrown in there.

**Response 14.** The paragraph is removed, and four references are taken away from the list of references.

**Comment 15.** It would be helpful to have a bit more of the big picture to better contextualize the work. Here is a suggested path for reworking the introduction:

Start with the real-world issue: offshore wind O&M is costly – how costly? See NREL work for example – therefore want to reduce

How does O&M for offshore wind work? again, see ECN O&M calculator as a good high level overview of how offshore wind O&M works in practice

But, many of these models rely on time-series simulation of the failures and logistics where there are many underlying probabilistic / uncertain events both with component failures (of various types) and access issues (related to weather conditions)

Thus, this becomes a very challenging optimization under uncertainty problem

How have people tackled this before?

In general, operations research communities have looked at O&M optimization via several techniques

For wind specifically, optimization of O&M has been tackled in x,y, z ways

In this paper, we build on the state of the art with a new algorithm, NextPM, that does x,y,z and thus has potential for capturing a,b,c, additional realism of the offshore wind O&M optimization problem and/or improves the computational efficiency by leveraging d,e,f…

The paper is organized as follows…

**Response 15.** The Introduction is thoroughly revised.

**Comment 16.** Appendix A and B have only the formal optimization problem formulations – is it necessary to have them as two separate appendices? Or appendices at all? Personally I prefer to see the problem formulation in situ and up front as it is the central organizing basis for an optimization study

**Response 16.** Appendix B is removed, and Appendix A is presented as a new Section 3.4.

**Optimal rescheduling algorithm**

**Comment 17.** Before jumping to the description of the algorithm, consider describing it at a highlevel in plan language

**Response 17.** A flow diagram is added.

**Comment 18.** FYI: A key cost of offshore wind failures is not the component costs but the downtime and associated loss of energy production. Most of the O&M models mentioned above also calculate availability which can then translate directly to revenue losses

**Response 18.** Thank you for sharing this valuable information!

**Comment 19.** FYI: most models above also treat dt is a uncertain variable since it is highly weather dependent, bj and cj are also uncertain but models often treat them as deterministic – i.e. failures are probabilistic but the costs to repair them are deterministic

**Response 19.** Thank you for sharing this valuable information!

**Comment 20.** Line 85 – need to explain here in plain terms what lamda is

**Response 20.** See Response 28 below.

**Comment 21.** Again, I think a flow diagram would help here. Can you represent visually or in plan language what the optimization problem you are trying to solve is? Contextualize it with an example using a wind turbine? And contrast it with a baseline approach? It is still unclear what is novel here.

**Response 21.** We added a flow diagram and explained how the methods work.

**Comment 22.** I am not sure why section 2 is a stand-alone section. It seems to me it is part of the methods and I recommend section 2 and 3 be merged and made subsections of an overarching methods section

**Response 22.** We hope the Section 2 is more justified in the revised version of the paper.

**Comment 23.** It is still not clear what NextPM is doing by the end of section 2 nor NextOM

**Response 23.** See response 21.

**Optimal plan for next preventive maintenance**

**Comment 24.** Overall section 3 would benefit from more contextualization from the real world offshore wind O&M problem. Nothing about section 3 seems tied specifically to the offshore wind O&M optimization problem. – can you bring in real world examples to help contextualize the approach and demonstrate its uniqueness and value from an offshore wind problem-specific perspective?

**Response 24.** We believe we have addressed this issue in our case studies.

**Comment 25.** Line 101-103 – this is important and should be discussed in the introduction. Also, it should be explained WHY this approach is being taken. It is not self-evident.

**Response 25.** We explained more carefully the role and practical choice of the parameter r in Section 4.1.

**Comment 26.** Line 104: Explain what z and x are from a real-world perspective

**Response 26.** We have made it clearer.

**Comment 27.** Line 110: why? Explain why this approach is being taken

**Response 27.** This is explained in Section 2.

**Comment 28.** Line 142 and 152 – why is lambda introduced? Please explain more clearly

**Response 28.** More explanation of lambda is added around the line 152.

**Comment 29.** Lines 212-215 should be brought into the introduction. It should already be clear up front how this work will extend the state of the art (though details can remain in a later section)

**Response 29.** We have changed the introduction to address this comment.

**Comment 30.** Also, line 212 is the FIRST mention in all of section 2 and 3 of a wind turbine. Nothing about the work to this point is tied to the offshore wind problem specific context… this is a problem for publishing in WES and needs to be remedied before acceptance to publication (see earlier notes on same topic)

**Response 30.** We acknowledge that our paper is not tied to the offshore wind problem specific context. However, see Response 11.

**Numerical studies**

**Comment 31.** There are much more recent papers related to wind turbine failures rates and repair costs – see work from organizations mentioned above and also reporting from NREL, ORE Catapult, BVG Associates, and others

**Response 31.** Unfortunately, we could not find a later than \cite{tian2011condition} reference suitable for the current context.

**Comment 32.** Typical farm lifes are more like 25 to 30 or longer – lifetime from a financing perspective is often 20 but has been getting longer

**Response 32.** This parameter has been changed from 20 to 30 in our calculations.

**Comment 33.** Line 224 – explain this lambda value better– at least in a footnote if not in a paragraph. What computer simulations elsewhere and why not reported???

**Response 33.** Yes, we have replaced

"… based on the analysis of computer simulations which is not reported here."

by

"… based on \cite{gustavsson2014preventive}."

**Comment 34.** Line 223 and line 235 – the decision of the specification of r is not well explained

**Response 34.** Explained on line 245.

**Comment 35.** Line 239 – d= 5, 5 what?

**Response 35.** See Response 38.

**Comment 36.** Lines 240-245, for wind systems, components are designed for the full plant lifetime (in this case 20 years). Any component replacement prior to that is considered a premature failure and thus is typically in the bucket of unplanned maintenance. Condition monitoring can help detect components that will fail prematurely, which is where the approach in this paper would become relevant. But it is odd to say that PM replacement (for any major component is 43 months) this would be SUPER short from an industry perspective. Again, this entire work seems pretty unlinked to the wind specific O&M problem

**Response 36.** Yes, 43 months is what our algorithm has produced for the next PM time under the assumptions made for the study. With different input parameters the output would be different.

**Comment 37.** Figure 1 shows the components but neither of the two figures are really well explained. It would be good to separate these into 2 figures and clearly explain what each of them means

**Response 37.** Yes, we divided Figure 1 into two figures and added explanations.

**Comment 38.** Line 250- Mobilization cost of d = 10, 10 what?

**Response 38.** Yes, now we repeatedly mention in the text that the cost unit is 1000 USD.

**Comment 39.** Part A and B and C headers (line 252, line 264 and line 272 respectively), make these separate lines and not in paragraph text

**Response 39.** Done.

**Comment 40.** Explain better what you mean by winter versus summer start – does this make sense from a real world problem perspective?

**Response 40.** We have clarified the three different settings. As to the reviewer's question: "does this make sense from a real world problem perspective? "- we think we addressed this question by our finding that:

"in all of the seasonal settings, the proposed PM activities are scheduled for summer months (having lower mobilization costs)."

**Comment 41.** I recommend doing Part C first – this would be the baseline (all corrective no predictive maintenance) – part A introduces some level of PM and then part B considers the seasonal effects. Then, include the percent savings in each of table 2 and 3 and reference back to the CM analysis

**Response 41.** Good idea, done.

**Comment 42.** Need to explain lines 286 to 289 better. How can you do a fair comparison?

**Response 42.** We have added an extra sentence:

"To make a fair comparison, we characterize both approaches in terms of the time average maintenance costs."

**Comment 43.** Line 304 – 5 hours to solve what? The full optimization problem?

**Response 43.** Yes, we added "the full optimization problem".

**Comment 44.** The overall comparison of NextPM and PMSPIC seems incomplete – are there caveats to this? What are the key assumptions you are making that might be limiting the external validity of the work?

**Response 44.** We have clarified the principal differences using new Figures 1 and 2.

**Conclusions**

**Comment 45.** Consider revisiting the conclusions after the rest of the paper is reworked and better tied to the wind energy problem

**Response 45.** Done.

**Comment 46.** Do not mention preliminary results (not shown) on computational time… either show them or don't mention them – i.e. leave it as future work

**Response 46.** Done.

**Comment 47.** Also, what is the future work? where will this effort go? What will it take to make this actually useful for wind farm O&M planning?

**Response 47.** We briefly mention the future work.

**Comment 48.** Line 316-317 again is more like a marketing pitch

**Response 48.** Line 316-317 are changed.

---

## Author Response (AR4)

**Reply to the comments of the editors**

We totally agree with the editors that the final comment, cited below, is very important. To address this comment we made two changes in the article.

**Comment.** Your paper is now nearly accepted I would like to improve the discussion of comment36, as outlined by the associate editor: "One issue that still is not addressed and needs to be in particular is comment 36 / response 36. I Should have made my concern on this stronger as this was one of my biggest concerns around the whole study and its validity from a wind turbine domain perspective (again compared to the state of the art). If you look for example at the O&M Model from ECN you all see that a key distinction is that the failure rates for minor to major repairs to full component replacements are quite different. You need to heavily qualify the entire study and findings based on the lack of distinguish between repairs and replacements and to heavily qualify the overall results based on the input assumptions. This needs to be corrected before final publication." Please includes such aspects in your discussion. with such a more technical correction your paper will be accepted. Best Joachim Peinke (Chief editor)

**Change 1.** In the case study section, just before Section 4.3, we added

"The optimal times for the next PM activity have landed in the range between 43 and 50 months and seem to be quite short. This is explained by the particular choice of the model parameters presented in Table 1: under the assumption of independence between the lives of the four components, the average time until the first failure is slightly below 50 months. (Notice that in this case study, the estimated parameters for the components' life lengths are based on the data collected for wind turbines from year 1994 to 2004. For the modern wind turbines, the mean survival times will be longer.) Another important contributing factor is the assumption of low PM costs, with higher PM costs the optimal next PM activity would be scheduled at a later time. In the special case with equal PM and CM costs, the optimal solution is to forget about PM planning and fully rely the pure CM strategy. "

**Change 2.** In the conclusion section, in the very end of Section 5, we replaced

"In the future, we plan to use NextPM as a key module in a maintenance scheduling app for wind turbines, after adding a module for processing condition monitoring data."

by

"The notable limitation of our setting is that it neglects such important maintenance activities as inspections, minor and major repairs. By considering full replacements as the only kind of CM and PM activities allowed in the model, we were able to tame the mathematical challenge of the problem in hand. Still, even within this simplified model framework, our computational analysis may bring useful insights of more efficient PM planning depending on a few key parameters of a concrete wind farm. Our results should be viewed as a first promising step towards a much more sophisticated mathematical optimization model that would take into account available condition monitoring data and even recognizing the difference in the failure rates for minor repairs, major repairs, and component replacements."